# Synthesis of Bioactive Nickel Nanoparticles Using Bacterial Strains from an Antarctic Consortium

**DOI:** 10.3390/md22020089

**Published:** 2024-02-14

**Authors:** Joseph Amruthraj Nagoth, Maria Sindhura John, Kesava Priyan Ramasamy, Alessio Mancini, Marco Zannotti, Sara Piras, Rita Giovannetti, Lydia Rathnam, Cristina Miceli, Maria Chiara Biondini, Sandra Pucciarelli

**Affiliations:** 1School of Biosciences and Veterinary Medicine, University of Camerino, 62032 Camerino, Italy; josephamruthraj.nagoth@unicam.it (J.A.N.); mariasindhura.john@unicam.it (M.S.J.); kesava.ramasamy@unicam.it (K.P.R.); alessio.mancini@unicam.it (A.M.); cristina.miceli@unicam.it (C.M.); mariachiara.biondini@unicam.it (M.C.B.); 2Department of Dermatology, University of Texas Southwestern Medical Center, Dallas, TX 75390, USA; 3Department of Ecology and Environmental Science, Umeå University, 90187 Umeå, Sweden; 4Chemistry Interdisciplinary Project (ChIP), Chemistry Division, School of Science and Technology, University of Camerino, 62032 Camerino, Italy; marco.zannotti@unicam.it (M.Z.); sara.piras@unicam.it (S.P.); rita.giovannetti@unicam.it (R.G.); 5Department of Physics, Pondicherry University, Puducherry 605014, India; drlydia86@gmail.com

**Keywords:** nanomaterials, green synthesis, Antarctic bacteria, antimicrobial activity, nosocomial pathogens

## Abstract

Marine microorganisms have been demonstrated to be an important source for bioactive molecules. In this paper we report the synthesis of Ni nanoparticles (NiSNPs) used as reducing and capping agents for five bacterial strains isolated from an Antarctic marine consortium: *Marinomonas* sp. ef1, *Rhodococcus* sp. ef1, *Pseudomonas* sp. ef1, *Brevundimonas* sp. ef1, and *Bacillus* sp. ef1. The NiSNPs were characterized by Ultraviolet–visible (UV–vis) spectroscopy, Dynamic Light Scattering (DLS), Transmission Electron Microscopy (TEM), X-ray diffraction (XRD) and Fourier Transform Infrared (FTIR) spectroscopic analysis. The maximum absorbances in the UV–Vis spectra were in the range of 374 nm to 422 nm, corresponding to the Surface plasmon resonance (SPR) of Nickel. DLS revealed NiSNPs with sizes between 40 and 45 nm. All NiSNPs were polycrystalline with a face-centered cubic lattice, as revealed by XRD analyses. The NiSNPs zeta potential values were highly negative. TEM analysis showed that the NiSNPs were either spherical or rod shaped, well segregated, and with a size between 20 and 50 nm. The FTIR spectra revealed peaks of amino acid and protein binding to the NiSNPs. Finally, all the NiSNPs possess significant antimicrobial activity, which may play an important role in the management of infectious diseases affecting human health.

## 1. Introduction

Antarctic marine bacteria represent an invaluable source of novel bioactive molecules that can be exploited in several biotechnological applications, including pharmaceutics [1]. Nanotechnology is gaining importance as a discipline for innovative materials and applications. Nanoparticles (hereafter called NPs) are becoming the fundamental building blocks of nanotechnology. NPs are materials between 1 and 100 nm in diameter. Their small dimensions and high surface area-to-volume ratio enable them to display novel chemical and physical properties. Special interest has been devoted to metal NPs, which are being widely used as industrial catalysts, in chemical sensing devices, in medical applications, in cosmetics, and as microelectronics [2,3,4,5,6].

Nickel NPs (NiSNPs) can be used as an electrode layer in multilayer ceramic capacitors (MLCC) [7] and as an interlayer for brazing stainless steel at lower temperatures [8]. Recently, NiSNPs have been synthesized through many routes such as polyol [9,10], sonochemical [11], thermal decomposition [12], and microemulsion [13]. However, physico-chemical synthetic protocols take a long time and result in a wider size range due to the thermal gradients. Furthermore, physico-chemical approaches are expensive and may produce toxic waste that needs to be treated before being discharged into the environment.

To counteract these problems, there is an urgent need to find inexpensive and environmentally friendly nanomaterial synthesis protocols. In this regard, the green synthesis of metallic NPs offers an attractive and inexpensive alternative to conventional physical and chemical methods [14]. The choice of green sources for solvent, reducing agent, and stabilizing/capping agent are three main criteria which should be handled with care in the green synthesis approach. Over the past decade, the use of water as a solvent has been prevalent, although, in some studies, other solvents such as ethanol and methanol have been used [15]. Two distinct biogenic sources with the potential to be applied as bioreducing agents in the green synthesis of NPs are microorganisms (e.g., bacteria, fungi, yeasts, and algae) and plants [15].

In this paper, we report a green protocol for the synthesis of NiSNPs using bacteria isolated from a consortium associated with the psychrophilic Antarctic ciliate *Euplotes focardii* [16,17,18]. The bacteria have been shown to be able to produce metal NPs, including silver and copper NPs [19,20,21,22], beside additional molecules that can be used in several industrial applications [23,24,25]. This biosynthetic approach appears to be a cost-efficient and promising alternative to conventional methods of synthesizing water-soluble NiSNPs. Synthesized NiS nanocrystals were deeply characterized and examined for their antibacterial activity. To the best of our knowledge, this is the first report of NiSNPs synthesis using Antarctic bacteria and it will provide useful references for further studies on biogenic Ni-based NPs.

## 2. Results and Discussion

### 2.1. Bacterial Synthesis of Nickel (Ni) Nanoparticles

We observed the formation of pale green extracellular aggregates in the form of fine powder at the bottom of the flasks 24 h after the addition of 1 mM of NiSO_4_ to the bacterial cultures: *Marinomonas* sp., *Rhodococcus* sp., *Pseudomonas* sp., *Brevundimonas* sp., and *Bacillus* sp., (Appendix A). Being the first bacteria isolated from a consortium associated with the ciliate *E. focardii*, these were named with the suffix ef1. The deposits were collected and washed several times with double distilled water and dried at room temperature for further characterization (Appendix A). The formation of pale green powder on the bottom of the flask was also observed after 120 h of incubation of *Microbacterium* sp. MRS-1 in liquid nutrient medium containing 2000 ppm of NiSO_4_ and this powder was characterized as Ni nanoparticles [26]. The mechanism involved in the formation of these NPs may be due to biologically controlled and metabolism-dependent processes which involve nucleation and deposition of inorganic particles and production of extracellular nanomaterials by efflux mechanism [27]. The characteristics of the synthesized NPs are summarized in Appendix A.

Ultraviolet–Visible (UV–Vis) spectrum analysis of the resulting NiSNPs gave absorption peaks close to 425 nm (Figure 1). However, *Bacillus* sp. ef1 and *Brevundimonas* sp. ef1 spectra also show a shift of the peaks toward 650 (D) or 550 nm (E), that can be attributed to synthesized NPs of different size, as reported in [28].

### 2.2. X-ray Diffraction (XRD) and Fourier Transform Infrared (FTIR) Spectroscopic Analysis

All synthesized NPs were examined by X-ray diffraction (XRD) to determine the exact crystalline structure (Figure 2). All NPs show the typical XRD pattern of the nickel sulfide particles; more details are as follows:-The observed diffraction peaks of *Marinomonas* sp. ef1 (Figure 2A) were at 30.1°, 44.82° 53.5°, 62.6° and 73.1°, attributed to (100), (102), (110), (200), and (202) planes of NiSNPs (ICDD 01-075-0613).-The peaks from *Rhodococcus* sp. ef1 display distinct diffraction peaks at 31.21°, 34.9°, 47.92°, 53.25°, 63.11°, 73.27°, attributed to (100), (101), (102), (110), (200), and (202) planes of NiSNPs (JCPDS reference 010750613).-XRD pattern of *Pseudomonas* sp. ef1 synthesized NiS NPs showed peaks at 31.11°, 34.92°, 47.93°, 53.40°, 63.11° were attributed to (100), (101), (102), (110) and (200) planes of NiSNPs (JCPDS2-1280).-*Brevundimonas* sp. ef1 showed formation of nanocrystalline peaks at 31.21°, 34.9°, 47.92°, 53.25° attributed to (100), (101), (102), (110), (200) and (202) planes of NiSNPs (JCPDS card nos: 2-1280).

XRD absorption peaks for *Bacillus* sp. ef1 synthesized NiS NPs are at 31.21°, 34.9°, 47.92°, 53.25°, 63.11°, and 73.27° and correspond to (100), (101), (102), (110), (200) and (202) planes of NiSNPs. The XRD pattern matched well with JCPDS data for NiSNPs (card no.01-075-0613).

FTIR analysis was carried out to identify capping biomolecules of the NiSNPs synthesized by the Antarctic bacteria. The FTIR analysis evidenced the presence of N–H, C–N, C–H and O–H groups, which corresponds to the presence of metabolites and proteins surrounding the NiSNPs. The FTIR spectra results are described in detail in the Appendix A. Mallikarjuna et al. [29] reported that the carbonyl and hydroxyl groups from amino acid residues or proteins can strongly bind to metal NPs like capping agents and stabilize the NPs by preventing its agglomeration. Sathyavathi et al. [30] also suggested that the biological molecules exhibit a dual role of formation and stabilization of NPs in the aqueous medium.

### 2.3. Dynamic Light Scattering (DLS) Analysis of Bacterial NiSNPs

Dynamic light scattering (DLS) was performed to estimate the particle’s size and zeta potential. *Marinomonas* sp. ef1, *Rhodococcus* sp. ef1, *Pseudomonas* sp. ef1, *Brevundimonas* sp. ef1 and *Bacillus* sp. ef1 NiSNPs showed average diameters of 42.3 nm, 44.8 nm, 42.1 nm, 40.7 nm, and 40.5 nm, respectively (Figure 3).

Similarly, the zeta potential values of *Marinomonas* sp. ef1, *Rhodococcus* sp. ef1, *Pseudomonas* sp. ef1, *Brevundimonas* sp. ef1 and *Bacillus* sp. ef1 NiSNPs are −32.2 mV, −31.1 mV, −28.5 mV, −30.6 mV and −29.3 mV, respectively (Figure 4).

### 2.4. Transmission Electron Microscopy (TEM) Analysis

TEM images of the bacterial synthesized NiSNPs showed both rods and spherical shapes (Figure 5). The size range of the spherical and rod NiSNPs are reported in Table 1. Our results are in good agreement with previous studies by Govindasamy et al. [31] and Wang et al. [32] that reported varied-shaped (irregular polygonal, cylindrical, and spherical) NiSNPs and agglomeration, probably due to magnetic interaction and polymer adherence between the particles.

### 2.5. Antibacterial and Antifungal Activity of NiSNPs

The Kirby–Bauer disk diffusion test was performed to verify the antimicrobial activity of the synthesized NiSNPs. We tested one Gram-positive bacterium (*Staphylococcus aureus*), Gram-negative bacteria (*Escherichia coli*, *Klebsiella pneumoniae*, *Proteus mirabilis*, *Pseudomonas aeruginosa*, *Serratia marcescens*, *Citrobacter koseri*, *Acinetobacter baumanii*) and two *Candida* species, *C. albicans* and *C. parapsilosis*, particularly widespread among the human fungal pathogens (Figure 6).

Clear inhibition zones are visible around the NiSNPs disks for each pathogen tested, suggesting that the NiSNPs possessed antimicrobial activity against both bacteria and fungi. The results of the Kirby–Bauer disk diffusion are summarized in Appendix A.

We also estimated the minimum inhibitory concentration (MIC) and the minimum bactericidal/fungicidal concentration MBC/MFC. MIC is defined as the lowest concentration of the antibacterial agent to inhibit pathogens’ growth by serial dilution. As shown in Table 1 and Appendix A, the NiSNPs synthesized by all the Antarctic bacteria show MIC values ranging from 3.12 to 25 μg/mL against the Gram-negative bacteria, whereas the MIC values ranged from 12.5 to 25 μg/mL against Gram-positive bacteria. The MIC values against fungi ranged from 6.25 to 25 μg/mL. MBC and MFC are defined as the lowest concentration of antibacterial or antifungal agent needed to kill the pathogens (i.e., no growth observed on the MHA plate). In the study, MBC/MFC values were in the range of 6.25 to 25 μg/mL. The results are reported in detail in the Appendix A. From these results, it appears that NiSNPs prepared by these Antarctic bacteria are particularly efficient against pathogenic Gram-negative bacteria. However, these NiSNPs demonstrated antibiotic effects also against *Candida* and the Gram-positive *Staphylococcus aureus*. In general, NiSNPs synthesized from Antarctic bacteria appear more effective than those obtained from *Ocimum sanctum* leaf extract, that showed maximum antimicrobial activity against tested pathogens at concentrations of 50–100 μg/mL, almost double than the antimicrobial concentrations of the NiSNPs reported in this paper [33].

Drug-resistant pathogens have created serious concerns across the globe due to the limited choices in antibiotic treatment [34]. A source of new biomolecules with antimicrobial activity against drug-resistant pathogens is represented by metabolites produced by novel isolated bacteria [35]. Our study highlights an efficient strategy in obtaining bionanomaterials that can be used as antibiotics against many of these pathogens. The antimicrobial activity of nickel NPs most probably relies on the generation of reactive oxygen species (ROS) and release of nickel ions Ni(II). Diffusion and endocytosis of Ni NPs followed by NiNPs’ accumulation in the cell membrane alters membrane permeability and destroys membrane proteins. Furthermore, Ni NPs react with water-forming ROS that penetrate the cell membrane causing damage with subsequent leakage of cellular contents. Dissolution of Ni (II) ions and free radicals interrupt electron transport in the microbial cell resulting in cell death [33]. Gram-negative bacteria showed higher inhibition zones than the Gram-positive *Staphylococcus aureus*. Vanaja et al. [36] reported that Gram-positive bacteria have thick and chemically complex peptidoglycan in the cell wall. Therefore, NPs are not easily entered into the Gram-positive bacterial cell. By contrast, Gram-negative bacteria have a thin cell wall, and the NPs can easily enter bacterial cells and cause their antimicrobial effects.

## 3. Materials and Methods

### 3.1. Cultures and Chemicals

*Marinomonas* sp. ef1, *Rhodococcus* sp. ef1, *Brevundimonas* sp. ef1, *Pseudomonas* sp. ef1 and *Bacillus* sp. ef1 strains used in this work were isolated from a bacterial consortium associated with the Antarctic ciliate *E. focardii*. All strains were maintained on Luria–Bertani agar (LBA) Petri dishes (Tryptone 10 g/L, Yeast extract 5 g/L, NaCl 5 g/L) at 22 °C (optimum growth temperature). All chemicals used were of analytical grade and were purchased from Sigma Aldrich (Milan, Italy) and Carlo Erba (Milan, Italy). Culturing media were purchased from ITW reagents (Milan, Italy).

### 3.2. Biosynthesis of NiSNPs

Bacterial strain colonies were inoculated in 100 mL of LB broth individually from agar plates and incubated overnight at 22 °C on a rotatory shaker set at 200 rpm. The bacterial growth was monitored by measuring the optical density (O.D.) of the culture media at λ = 600 nm to reach an OD of about 2.8–2.9. At this stage the cells were in the mature and high dividing stage of their life cycle. Later, NiSO4 was added to the cultures to a final concentration of 1 mM. The cultures were then incubated in bright conditions for 24 h on a rotator shaker at 150 rpm and 22 °C to facilitate the biosynthesis of NPs. As a control, 24 h culture media (Luria–Bertani) with 1 mM NiSO_4_ without the organisms and heat killed bacterial cultureswere maintained. The preliminary detection of NPs’ synthesis was carried out by visual observation of a color change of the reaction mixture. The synthesis was then confirmed by UV–Vis spectra using a Shimadzu UV 1800 spectrophotometer.

### 3.3. Extraction and Purification of NiSNPs

NiSNPs were collected from both intracellular and extracellular fractions. Briefly, 5 mL culture samples were collected in a sterile 15 mL Falcon tube individually and centrifuged at 7000× *g* on a Beckmann J2-21 (Beckmann Coulter, Milan, Italy), at 4 °C for 10 min, to separate the extracellular fraction and the bacterial pellet. To remove any uncoordinated biological molecules, the supernatant containing the extracellular fraction of the NPs was again centrifuged at 17,000× *g* for 15 min, and the NPs recovered in the supernatant. Furthermore, the bacterial pellet was suspended in 4 mL of sterile deionized water and stored at −20 °C in a deep freezer. The samples were frozen and thawed four times and vortexed to brake bacteria and to extract the NPs. The samples were centrifuged at 800× *g*, at 4 °C for 10 min. The resulting pellet was discarded, and the supernatant was used as the intracellular fraction NPs sample. Intra- and extracellular NPs were washed twice with deionized water by centrifugation steps at 16,000× *g*, and dried. The recovered NiS NPs were used for further characterization and antibacterial experiments.

### 3.4. Characterization of NiSNPs

The absorption optical spectrum of NiSNPs was recorded using Shimadzu UV-2401 PC double beam spectrophotometer from the range between 300 and 600 nm at (25 ± 1) °C. The particle size distribution and zeta potential of the NPs were studied using Zetasizer Nano-ZS90 (Malvern Instruments, Worcestershire, UK). TEM analysis was performed using JOEL Model 1200 FX under 80 kV power supply. Samples were prepared by placing drops of NiSNPs suspension over a carbon coated copper grid, extra solution was removed by blotting paper and the solvent was allowed to evaporate with the help of a critical point dryer. An image was obtained by the interaction of electrons transmitted through the specimen. The functional groups capped on the surface of the NiSNPs were identified with the FTIR spectrum using the Nicolet380 FTIR spectrometer. The sample was prepared and stabilized under reactive humidity before acquiring the spectrum. The spectrum was measured between 400 and 4000 cm^−1^ at the resolution of 4 cm^−1^. The X-ray diffraction (XRD) measurements were performed using a Rigaku-D/MAX-PC 2500 X-ray diffractometer (Rigaku, Milan, Italy) with a Cu Kα (λ = 1.5405 Å) in 2θ range from 0 to 80° at scan rate of 0.03° S^−1^.

### 3.5. Assessment of Antibacterial Properties

#### 3.5.1. Bacteria Strains Preparation

The antibacterial activity of biosynthesized NiSNPs was tested against *Staphylococcus aureus* (ATCCV@25923), *Escherichia coli* (ATCCV@25922), *Klebsiella pneumoniae* (ATCCV@13883), *Pseudomonas aeruginosa* (ATCCV@27853), *Proteus mirabilis* (ATCCV@35659), *Citrobacter koseri* (ATCCV@25408), *Acinetobacter baumanii* (ATCCV@19606), *Serratia marcescens* (ATCCV@13880), *Candida albicans* (ATCCV@ 90028) and *Candida parapsilosis* (ATCCV@22019). All the strains were cultured in Mueller Hinton broth (MHB) (Merck, Germany) at 37 °C for 24 h with 200 rpm agitation.

#### 3.5.2. In Vitro Antibacterial Susceptibility Test

##### Kirby–Bauer Disk Diffusion Susceptibility Test

The antibacterial activity of NiSNPs against the selected bacterial strains was carried out using Kirby–Bauer Disk Diffusion Susceptibility Test method. Using sterile cotton swabs, the bacteria strains were spread on the Mueller-Hinton agar (MHA). The disks were loaded with 25 μL (25 μg) of NiS NPs solution and Ni sulphate solution (1 mM) and dried. The disks were then placed on the agar plate and incubated at 37 °C for 24 h. The zone of inhibition was observed after 24 h of incubation.

##### Minimum Inhibitory Concentration (MIC), Minimum Bactericidal (MBC) and Minimum Fungicidal (MFC) Concentration Evaluation

The MIC and MBC/MFC evaluations of the green synthesized NiSNPs were carried out using the method described in the guideline of CLSI 2012 [19]. The MIC test was performed in a 96-well round bottom microtiter plate using standard broth microdilution methods while the MBC/MFC test was performed on the MHA plates. The bacterial inoculums were adjusted to the concentration of 0.5 McFarland units. For the MIC test, NiSNPs stock solution was prepared by dispersing it in sterilized deionized water using ultrasonic to reach 200 μg/mL. A total of 100 μL of stock solution was serially diluted twofold in 100 μL of MHB in the first row and finally 100 μL was discarded such that the first well in the row of the microtiter plate contained the highest concentration of NiSNPs, while the last well of the row contained the lowest concentration. Similarly, NiSNPs were prepared in all the rows. Positive control contains only medium (K+) and negative control contains medium and bacterial inoculums (K−). The microtiter plate was then incubated at 37 °C for 24 h. The MIC value is defined as the lowest concentration of antibacterial agent that inhibits the growth of bacteria. The MBC/MFC values were taken as the lowest concentration of antimicrobial agents that completely kill the pathogens. To check MBC/MFC, the suspensions from each well of the microtiter plates were spread into MHA plates and were incubated at 37 °C for 24 h. MBC value was taken as the lowest concentration with no visible growths on the MHA plates.

## 4. Conclusions

NiSNPs were successfully produced from Antarctic bacteria. The methodology is simple and energy efficient. This approach shortens the reaction time and makes it easy to continuously synthesize large quantities of products for industrial application. NPs were characterized by using various analytical techniques such as UV–Vis absorption spectroscopy, DLS, Zeta potential, FTIR, and XRD. The characterization studies confirmed the formation of NiSNPs using the green synthesis. TEM analysis showed varied-shaped NiSNPs in good agreement with previous reports. The synthesized NiSNPs had the most effective antimicrobial activity against pathogenic Gram-negative bacteria but demonstrated antibiotic effects also against the two tested *Candida* species and the Gram-positive *Staphylococcus aureus*. This activity is higher than that reported for plant-synthesized NPs. Antarctic bacteria have again been proven to be a precious source of bioactive material and they will provide useful references for further studies on this subject.

## Figures and Tables

**Figure 1 marinedrugs-22-00089-f001:**
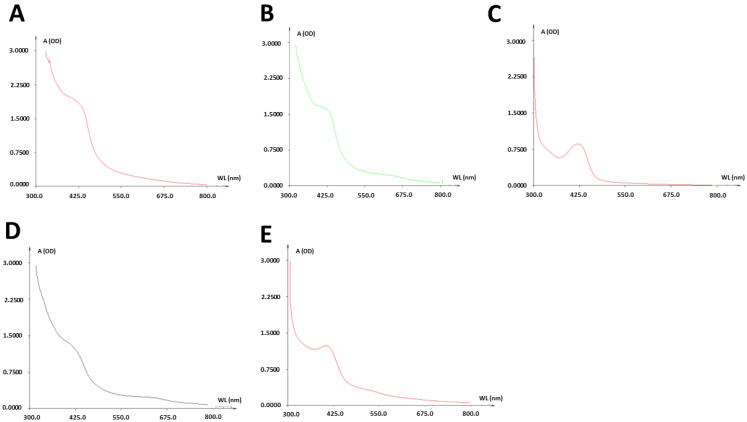
UV–Vis absorbance spectra of NiSNPs synthesized from (**A**) *Marinomonas* sp. ef1, (**B**) *Rhodococcus* sp. ef1, (**C**) *Pseudomonas* sp. ef1, (**D**) *Brevundimonas* sp. ef1, and (**E**) *Bacillus* sp. ef1.

**Figure 2 marinedrugs-22-00089-f002:**
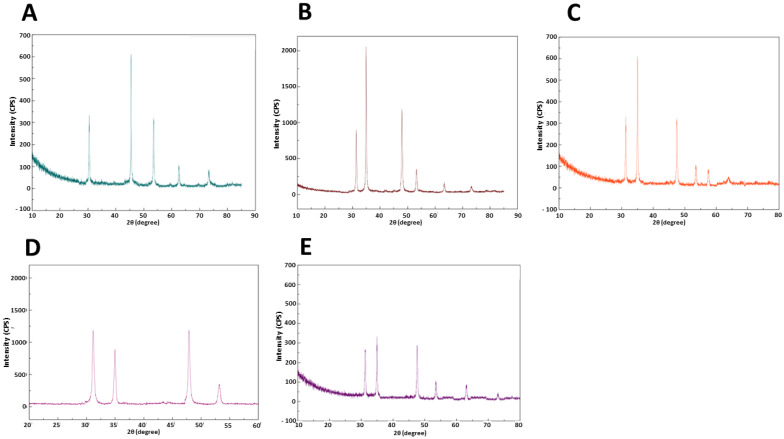
X-ray diffraction profiles of NiSNPs synthesized from (**A**) *Marinomonas* sp. ef1, (**B**) *Rhodococcus* sp. ef1, (**C**) *Pseudomonas* sp. ef1, (**D**) *Brevundimonas* sp. ef1, and (**E**) *Bacillus* sp. ef1.

**Figure 3 marinedrugs-22-00089-f003:**
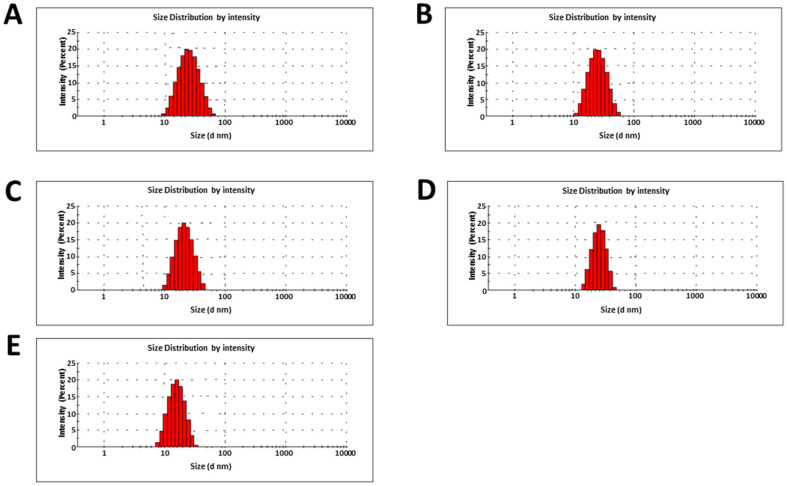
Dynamic light scattering (DLS) profiles of NiSNPs synthesized from (**A**) *Marinomonas* sp. ef1, (**B**) *Rhodococcus* sp. ef1, (**C**) *Pseudomonas* sp. ef1, (**D**) *Brevundimonas* sp. ef1, and (**E**) *Bacillus* sp. ef1.

**Figure 4 marinedrugs-22-00089-f004:**
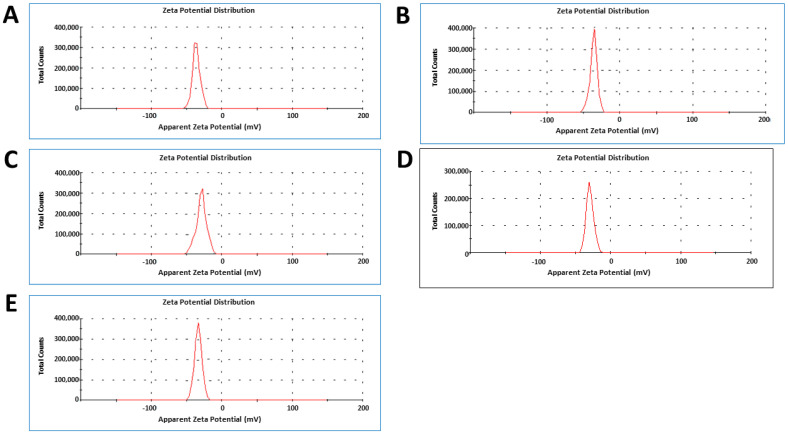
Zeta potential values of NiSNPs synthesized from (**A**) *Marinomonas* sp. ef1, (**B**) *Rhodococcus* sp. ef1, (**C**) *Pseudomonas* sp. ef1, (**D**) *Brevundimonas* sp. ef1, and (**E**) *Bacillus* sp. ef1.

**Figure 5 marinedrugs-22-00089-f005:**
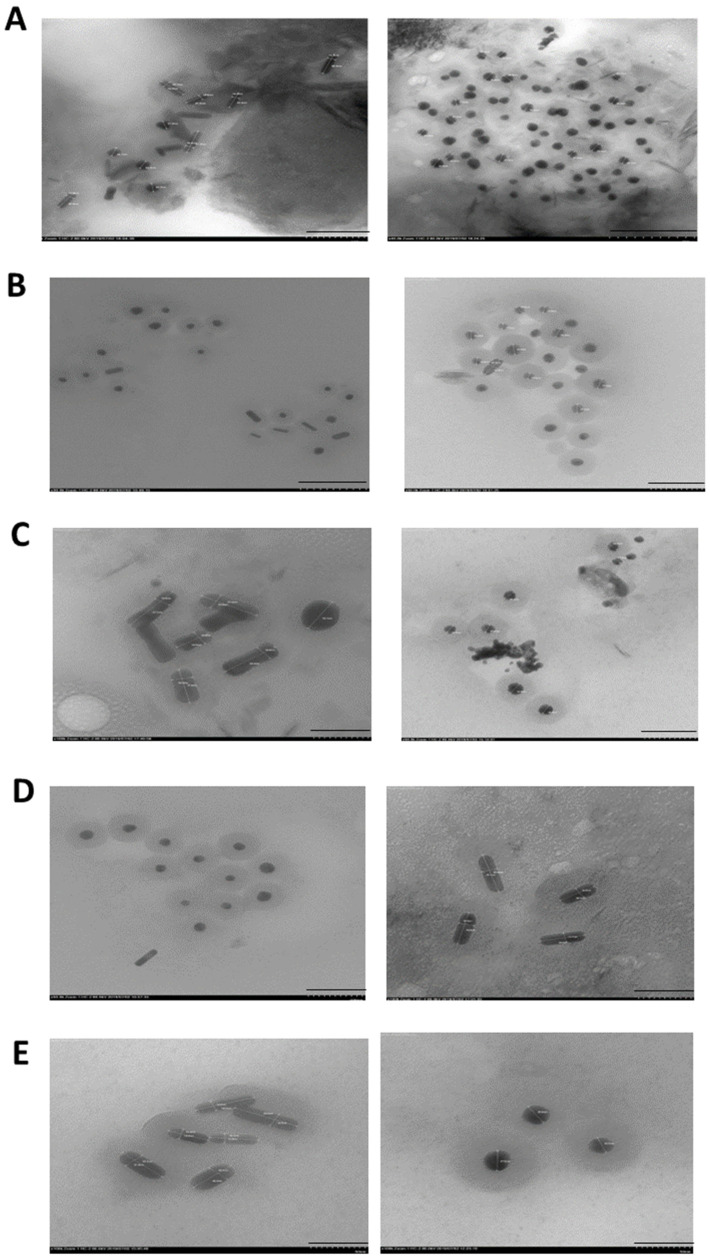
Transmission Electron Microscopy (TEM) of NiSNPs synthesized from the Antarctic bacteria. The bars correspond to 100 nm. In white, the measurement of NP diameters, reported as follow: (**A**) *Marinomonas* sp. ef1 NPs: the average size is 40–50 nm, for the rod shaped; 20–30 nm for the spherical shaped (**B**) *Rhodococcus* sp. ef1 NPs: the average size is 25–50 nm for the rod shapes and 15–30 nm for the spherical shaped; (**C**) *Pseudomonas* sp. ef1 NPs: the average size is 30–50 nm, for the rod shaped; 20–30 nm for the spherical shaped, (**D**) *Brevundimonas* sp. ef1 NPs: the average size is 40–50 nm, for the rod shaped; 20–30 nm for the spherical shaped, and (**E**) *Bacillus* sp. ef1 NPs: the average size is 30–50 nm, for the rod shaped; 20–30 nm for the spherical shaped.

**Figure 6 marinedrugs-22-00089-f006:**
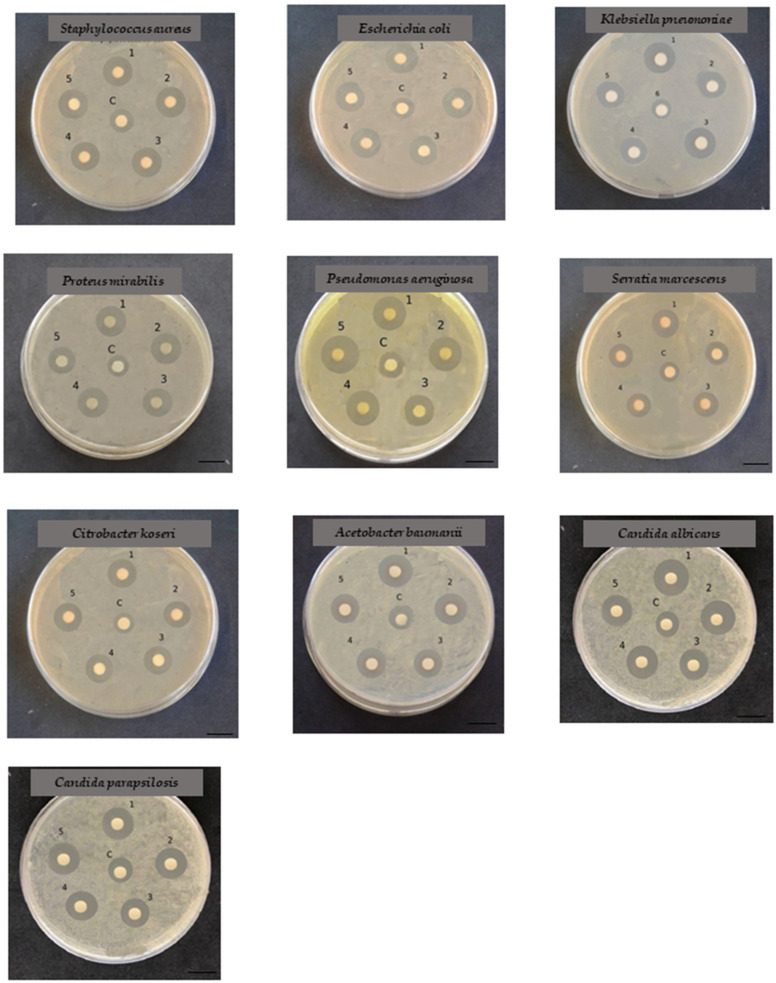
Antibacterial and antifungal activity of NiSNPS by Kirby–Bauer Disk Diffusion Susceptibility Test against pathogenic and Gram-positive bacterium (*Staphylococcus aureus*), Gram-negative bacteria (*Escherichia coli*, *Klebsiella pneumoniae*, *Proteus mirabilis*, *Pseudomonas aeruginosa*, *Serratia marcescens*, *Citrobacter koseri*, *Acinetobacter baumanii*) and two *Candida* species, *C. albicans* and *C. parapsilosis*. 1. *Marinomonas* sp. ef1 synthesized NiSNPS; 2. *Rhodococcus* sp. ef1 synthesized NiSNPS; 3. *Brevundimonas* sp. ef1 synthesized NiSNPS; 4. *Pseudomonas* sp. ef1 synthesized NiSNPS; 5. *Bacillus* sp. ef1 synthesized bio NiSNPS. C: control (1 mM NiSO_4_). Bars: 10 mm.

**Table 1 marinedrugs-22-00089-t001:** MIC and MBC/MFC values (µg/mL) of the bio NiSNPs against pathogens.

	*Marinomonas*		*Rhodococcus*		*Brevundimonas*		*Pseudomonas*		*Bacillus*	
	MIC	MBC	MIC	MBC	MIC	MBC	MIC	MBC	MIC	MBC
*Staphylococcus aureus*	12.5 ± 0.2	25.0 ± 0.4	25.0 ± 0.4	25.0 ± 0.4	12.5 ± 0.4	25.0 ± 0.4	25.0 ± 0.4	25.0 ± 0.4	12.5 ± 0.4	12.5 ± 0.4
*Escherichia coli*	25 ± 0.4	25 ± 0.3	12.5 ± 0.1	25 ± 0.4	25 ± 0.5	25 ± 0.2	25 ± 0.2	25 ± 0.3	12.5 ± 0.1	12.5 ± 0.4
*Klebsiella pneumoniae*	12.5 ± 0.2	25 ± 0.2	12.5 ± 0.2	25 ± 0.5	12.5 ± 0.3	25 ± 0.4	25 ± 0.4	25 ± 0.4	6.25 ± 0.1	12.5 ± 0.3
*Pseudomonas aeruginosa*	12.5 ± 0.1	12.5 ± 0.2	12.5 ± 0.3	25 ± 0.43	12.5 ± 0.4	25 ± 0.2	12.5 ± 0.2	12.5 ± 0.4	12.5 ± 0.2	12.5 ± 0.4
*Proteus mirabilis*	3.12 ± 0.2	6.25 ± 0.4	6.25 ± 0.1	12.5 ± 0.4	12.5 ± 0.2	12.5 ± 0.1	6.25 ± 0.2	12.5 ± 0.3	6.25 ± 0.4	12.5 ± 0.4
*Citrobacter koseri*	6.25 ± 0.2	12.5 ± 0.4	12.5 ± 0.2	12.5 ± 0.3	12.50.4	25 ± 0.2	6.25 ± 0.1	12.5 ± 0.4	12.5 ± 0.2	12.5 ± 0.3
*Acinetobacter baumanii*	12.5 ± 0.3	12.5 ± 0.2	6.25 ± 0.3	12.5 ± 0.2	12.5 ± 0.3	25 ± 0.4	12.5 ± 0.3	12.5 ± 0.5	12.5 ± 0.2	12.5 ± 0.1
*Serratia marcescens*	6.25 ± 0.2	12.5 ± 0.4	6.25 ± 0.4	12.5 ± 0.2	6.25 ± 0.1	12.5 ± 0.3	12.5 ± 0.4	12.5 ± 0.2	6.25 ± 0.1	12.5 ± 0.4
	MIC	MFC	MIC	MFC	MIC	MFC	MIC	MFC	MIC	MFC
*Candida albicans*	25 ± 0.4	25 ± 0.3	12.5 ± 0.5	25 ± 0.1	12.5 ± 0.4	25 ± 0.4	25 ± 0.2	25 ± 0.4	12.5 ± 0.4	25 ± 0.2
*Candida parapsilosis*	12.5 ± 0.1	25 ± 0.4	25 ± 0.3	25 ± 0.2	12.5 ± 0.2	25 ± 0.3	12.5 ± 0.1	25 ± 0.3	6.25 ± 0.2	12.5 ± 0.3

## Data Availability

Data are contained within the article.

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
