# Peer review of "Synthesis of Bioactive Nickel Nanoparticles Using Bacterial Strains from an Antarctic Consortium"

_marinedrugs, 2024, doi:10.3390/md22020089_

Round 1

Reviewer 1 Report

Comments and Suggestions for Authors

The work concerns novel approach for the synthesis of Ni nanoparticles using marine bacteria with antimicrobial potential. Since nanoparticles are one of the solution to combat drug-resistant strains it appears to be an important issue. The results obtained by authors showed a success in the production of NiSNPs using five bacterial strains. Also the antimicrobial tests showed relatively low MIC values, especially against Gr- bacteria.

Although the paper is generally well written I have raised some issues that need to be addressed:

-Authors tested antimicrobial activity against one Gr+ species, few Gr- and two yeasts: Candida albicans and Candida parapsilosis that belong to Fungi. However, authors describe this tests as "antibacterial" - please correct in appropriate places in the manuscript to "antibacterial and antifungal" or "antimicrobial" (e.g., Fig 6 legend). Also there is no MBC for fungi, but MFC (minimum fungicidal concentration)

- Please, give a comment on why these two particular yeast species were used.

- In results and discussion please address the results obtained on yeasts. Fungi generally have a thick cell wall that contains chitin (among others), so are the results more similar to those for Gr+ bacteria?

- what exactly are the controls used for antimicrobial tests? As I understood authors treated bacteria and fungi incubated without NPs as a positive control, but the positive control would be the test with NPs with documented antimicrobial activity so we would be able to compare the effects. The sample without the antimicrobial compound is more of a growth control. Also a blank control should be used (perhaps it was) to subtract the starting inoculum.

- please include positive control (with documented NP, e.g., NiNPs synthesized traditionally) in the table with MIC values

- In my opinion MIC and MBC/MFC values have more power than inhibition zones (if MBC/MFC is studied it shows if the tested compound is bacteri/fungicidal or static) so I would recommend  to include those tables in the manuscript and inhibition zones in supplementary materials

- For antibacterial testing MH is recommended medium, but for yeasts is RPMI, YNB or Sab. Please, give a comment on the choice of medium for yeasts.

- Please indicate the source of strains for antimicrobial tests - are those strains clinical or reference? Especially, in the case of Pseudomonas sp since the species is not given in the manuscript. Among Pseudomonas sp P. aeruginosa is the most commonly tested for antibacterial compounds, since it posses multiple mechanisms of drug-resistance.

Author Response

Reviewer 1

The work concerns novel approach for the synthesis of Ni nanoparticles using marine bacteria with antimicrobial potential. Since nanoparticles are one of the solution to combat drug-resistant strains it appears to be an important issue. The results obtained by authors showed a success in the production of NiSNPs using five bacterial strains. Also the antimicrobial tests showed relatively low MIC values, especially against Gr- bacteria.

Although the paper is generally well written I have raised some issues that need to be addressed:

-Authors tested antimicrobial activity against one Gr+ species, few Gr- and two yeasts: Candida albicans and Candida parapsilosis that belong to Fungi. However, authors describe this tests as "antibacterial" - please correct in appropriate places in the manuscript to "antibacterial and antifungal" or "antimicrobial" (e.g., Fig 6 legend). Also there is no MBC for fungi, but MFC (minimum fungicidal concentration).

Response: We corrected as suggested in the revised version of the paper.

- Please, give a comment on why these two particular yeast species were used.

Response: in the revised version of the paper, we added a comment on why the two Candida species were used, explaining that these are among the most widespread in the hospitals (lanes 158-159).

- In results and discussion please address the results obtained on yeasts. Fungi generally have a thick cell wall that contains chitin (among others), so are the results more similar to those for Gr+ bacteria?

Response: unfortunately, the mechanisms of nanoparticle activity as antimicrobial are not known. Therefore, we cannot make any comments about this result. We only added a phrase enphatazing the higher activity on gram negative bacteria (lane 183).

- what exactly are the controls used for antimicrobial tests? As I understood authors treated bacteria and fungi incubated without NPs as a positive control, but the positive control would be the test with NPs with documented antimicrobial activity so we would be able to compare the effects. The sample without the antimicrobial compound is more of a growth control. Also a blank control should be used (perhaps it was) to subtract the starting inoculum.

Response: we added this information in the table S2.

- please include positive control (with documented NP, e.g., NiNPs synthesized traditionally) in the table with MIC values

Response: We cannot compare with traditionally synthesized NiNPs since our nanoparticles are NiSNPs and possess different capping properties 

- In my opinion MIC and MBC/MFC values have more power than inhibition zones (if MBC/MFC is studied it shows if the tested compound is bacteri/fungicidal or static) so I would recommend  to include those tables in the manuscript and inhibition zones in supplementary materials

Response: moved this table in the main manuscrit (new table 1)

- For antibacterial testing MH is recommended medium, but for yeasts is RPMI, YNB or Sab. Please, give a comment on the choice of medium for yeasts.

Response: we used MH agar to compare the results from fungi with those from bacteria

- Please indicate the source of strains for antimicrobial tests - are those strains clinical or reference? Especially, in the case of Pseudomonas sp since the species is not given in the manuscript. Among Pseudomonas sp P. aeruginosa is the most commonly tested for antibacterial compounds, since it posses multiple mechanisms of drug-resistance.

Response: we added this information under material and methods (lanes 265-271)

Reviewer 2 Report

Comments and Suggestions for Authors

1. Line 3, “.” at the end of title should be removed.

2. Line 57-59, the sentence should have reference.

3. In figure 1, 2, 3 and 4, the number and words in the ordinate and the abscissa are not clear.

4. the peak in figure 1D and figure 1E is not according with the present in texts.

5. Line 115, 153, and 184-186, “ Ni NPs” should be “NiSNPs”.

6. Line 116, Antarctic bacteria should be “ Antarctic bacteria”.

7. Line 145, “I” should be 1.

8. Line 155, 162, 165-167, 218, 228, 247, 257 and 287, and in the title of Table S1 and Figure S1, S2 and S3, “NiS NPs” should be “NiSNPs”.

9. In figure 6 and line 248-250, the name of bacteria should be italic (type) .

10. In figure 6, the negative control was less.

11. Line 163-165, the name of bacteria may be removed because they were presented in figure.

12. Line 185 and table S2, “ NiNPs” should be “NiSNPs”.

13. Line 192, “ the cell” may be “the gram positive bacteria cell”.

14. The control materials in KB test was Zinc sulphate in line 257, but it was NiSO4 in line 168.

15. Line 288, it refer to unknown。

16. In the title of table S2, the 4 in “NiSO4” should be subscript, and “ negative control” may be positive control.

17. The name of NiSNPs in papers should be the same with the table S2 and S3.

18. Line 241, “ cm−1” should be “ cm−1 , which may be agreement with Supplementary results. And line 244, “S−1” may be “S−1 .

Comments on the Quality of English Language

It could be better.

Author Response

Reviewer 2

  1. Line 3, “.” at the end of title should be removed.

Response: In the revised version of the paper, we removed the dot at the end of title

  1. Line 57-59, the sentence should have reference.

Response: in the revised version of the paper, we added the reference (lane 59 in red).

  1. In figure 1, 2, 3 and 4, the number and words in the ordinate and the abscissa are not clear.

Response: in the revised version of the paper we improved the quality of the figures.

  1. the peak in figure 1D and figure 1E is not according with the present in texts.

Response: in the revised version of the paper we change the description of these peaks (line 91, in red).

  1. Line 115, 153, and 184-186, “ Ni NPs” should be “NiSNPs”.

Response: in the revised version of the paper we changes all the Ni NPs” into  “NiSNPs”.

  1. Line 116, “ Antarctic bacteria” should be “ Antarctic bacteria”.

Response: done

  1. Line 145, “I” should be “1”.

Response: done

  1. Line 155, 162, 165-167, 218, 228, 247, 257 and 287, and in the title of Table S1 and Figure S1, S2 and S3, “NiS NPs” should be “NiSNPs”.

Response: done

  1. In figure 6 and line 248-250, the name of bacteria should be italic (type) .

Response: we corrected all names in italics but we removed the bacteria names in figure 2.

  1. In figure 6, the negative control was less.

Response: in fig 6 we used as a control the NiSO4 salt to verify that the antimicrobial effect was due to the nanoparticles and not of the salt

  1. Line 163-165, the name of bacteria may be removed because they were presented in figure.

Response: we wish to mantain the name of bacteria to better describe the species

  1. Line 185 and table S2, “ NiNPs” should be “NiSNPs”.

Response: in the revised version of the paper we changes all the Ni NPs” into  “NiSNPs”.

  1. Line 192, “ the cell” may be “the gram positive bacteria cell”.

Response: done

  1. The control materials in KB test was Zinc sulphate in line 257, but it was NiSO4 in line 168.

Response: in the revised version of the paper Zinc was corrected with Ni

  1. Line 288, “it” refer to unknown。

Response: in the revised version of the paper we corrected the sentence. Line 305

  1. In the title of table S2, the “4” in “NiSO4” should be subscript, and “ negative control” may be “positive control”.

Response: We changed NiSO4 into NiSO4. We prefer to label the NiSO4 reaction as “a control to verify the effect of nanoparticles and not of the salt”.

  1. The name of NiSNPs in papers should be the same with the table S2 and S3.

Response: ok

  1. Line 241, “ cm−1” should be “ cm−1” , which may be agreement with Supplementary results. And line 244, “S−1” may be “S−1” .

Response: done